# Basic In-Mouth Attribute Evaluation: A Comparison of Two Panels

**DOI:** 10.3390/foods8010003

**Published:** 2018-12-21

**Authors:** Mihaela Mihnea, José Luis Aleixandre-Tudó, Martin Kidd, Wessel du Toit

**Affiliations:** 1Department of Viticulture and Oenology, Stellenbosch University, ZA-7600 Stellenbosch, South Africa; joaltu@sun.ac.za (J.L.A.-T.); wdutoit@sun.ac.za (W.d.T.); 2RISE Research Institutes of Sweden, Agrifood and Bioscience, P.O. Box 5401, SE-402 29 Gothenburg, Sweden; 3Department of Statistics and Actuarial Sciences, Stellenbosch University, ZA-7600 Stellenbosch, South Africa; mkidd@sun.ac.za

**Keywords:** astringency, comparison, red wine, trained panel, winemakers

## Abstract

Astringency is often difficult to evaluate accurately in wine because of its complexity. This accuracy can improve through training sessions, but it can be time-consuming and expensive. A way to reduce these costs can be the use of wine experts, who are known to be reliable evaluators. Therefore, the aim of this work was to compare the sensory results and the panel performance obtained using trained panelists versus wine experts (winemakers). Judges evaluated twelve red wines for in-mouth basic perception (sweet, sour, bitter, astringent, and burning sensation) following the same tasting protocol and with the samples being presented in two different tasting modalities. Panels’ performance and relationship between the chemical composition and the sensory perception were investigated. Both panels showed similar consistency and repeatability, and they were able to accurately measure the astringency of the wines. However, the significant correlations between sensory scores and chemical composition varied with the panel and the tasting modality. From our results, we could see that winemakers tended to discriminate better between the samples when the differences were very small.

## 1. Introduction

Mouthfeel plays an important role in the quality of red wines [1]. It includes basic taste and other sensations, such as temperature, burning, body, prickling, and astringency. Astringency is probably the most important of these in-mouth sensations, as it contributes to the complexity and quality of red wines [2,3,4]. This attribute has been defined as “the complex of sensations due to shrinking, drawing, or puckering of the epithelium as a result of exposure to substances such as alums or tannins” [5]. It is described as a combination of different feelings, especially drying and roughness [6], and even though most people are familiar with it, it is often difficult to evaluate. 

It is generally accepted that phenolic compounds, especially tannins, are the main contributors to the perception of astringency in red wine [7,8]. Different authors [9,10,11,12,13] have shown positive correlations between astringency intensities and tannin levels in wine. However, the perceived intensity of the astringency varies not only with the type of tannins but also with the wine matrix: Ethanol, anthocyanin, and polysaccharide levels have also been shown to play an important role when studied in model wine [14,15]. 

The ability to objectively measure wine quality, in which wine tannins and astringency play a crucial role [16], is important for wine producers. Several precipitation-based methods to quantify tannin content have been developed in the past years [17,18,19]. Ovalbumin, saliva, bovine serum albumin or methylcellulose are some of the reagents used to react and precipitate tannins. Moreover, some studies have reported good correlations between tannin levels measured through the methyl cellulose precipitable (MCP) method and the perceived astringency when using trained sensorial panels [20,21,22].

The possibility to estimate tannin levels through simple measurements is important, especially for winemakers, as it could help them in everyday decision making [23] and the use of IR-spectroscopy techniques are considered to be cost effective, timeless and reliable approaches to achieving this aim [24,25]. 

However, predicting sensory perceptions is a difficult task, as it involves humans, which can induce to large in-data variations due to their unique physiological characteristics and individual differences. The only way to reduce this variation is by training the sensory assessors or by using wine experts. It is a complex and difficult task to measure the intensity of astringency induced by tannins, due to confusion between sourness, bitterness and astringency [8,26]. Research into the physiology of taste demonstrates the difficulty of gaining expertise in wine evaluation [27]. 

In order to accurately distinguish between these attributes [28], it is often necessary to train a panel, which usually translates into a time-consuming and expensive process. An accurately trained panel requires panelists with high consistency and consensus [29]. Consistency refers to the ability of panelists to accurately repeat the evaluation of the same product, whereas consensus refers to the similarity of the evaluation of the same product between judges tasting independently. According to ASTM (American Society for Testing and Materials) [5], a trained panelist is a person with a high degree of sensory acuity who has experience in the test procedure and an established ability to perform consistent and repeatable sensory assessments. An expert is a person who has extensive experience in a product category and who performs perceptual evaluations and who can draw conclusions about the effects of variations in raw materials or aging. Bearing this in mind, the use of wine experts instead of trained panelists could be an alternative to reducing the costs involved in these types of sensory studies.

Various researchers have suggested that experts can adopt analytical strategies [30,31,32,33]. Generally, experts’ assessments correlate well with trained panelists’ ratings when evaluating quality or concepts [34,35,36,37,38]. Moreover, when both a trained panel and experts perform the same task under the exact same conditions, various studies have shown that the experts’ performance is similar to that of the trained panelists, and sometimes even better.

Considering the ASTM definitions and the literature that shows that experts may adopt an analytical strategy similar to trained panels, our hypothesis was based on the idea that experts and a trained panel would give similar sensory responses when evaluating in-mouth basic sensations for the same wines. It was also expected that the possible correlations between the chemical composition and the sensory perceptions of the wines could be similar between panels. 

The aim of the present study was thus to compare the performance of wine experts and a trained panel when assessing in-mouth perception, using the same wine samples and tasting protocol. The two panels were compared by assessing consistency and consensus among judges and panels, correlations between attributes, correlations between attributes of the product and the chemistry, and similarity in wine discrimination. Grape variety and tasting modality effects were also assessed.

## 2. Materials and Methods 

### 2.1. Chemicals and Standards

Hydrochloric acid (HCl), gallic acid, catechin, p-coumaric acid, quercetin-3-glucoside, and quercetin were obtained from Sigma-Aldrich Chemie (Steinheim, Germany). In addition, phosphoric acid and caffeic acid were purchased from Fluka (Sigma-Aldrich Chemie, Steinheim, Germany). Acetonitrile was obtained from Merck (Darmstadt, Germany), and malvidine-3-glucoside chloride was purchased from Extrasynthese (Lyon, France). All chemicals and standards were HPLC (high-performance liquid chromatography) grade.

### 2.2. Wine Samples

For this study, 12 different South African red wines with different MCP tannin values were chosen, expected to cover different ranges of astringency intensities. To this end, 52 different Cabernet Sauvignon, Pinotage, and Shiraz red wines from 2015, originating from different wineries in the Western Cape region, were screened for their phenolic compounds composition, including tannins. With the use of multivariate tools such as principal component analysis (PCA), a total of 12 different wines, 4 wines from each variety, were chosen, ranging from higher to lowest tannin content.

### 2.3. Wine Characterization

The selected wines were chemically characterized evaluating their different oenological parameters and their phenolic composition. MCP tannin levels were predicted using the models constructed by Aleixandre-Tudo et al. (2015) [23]. The root mean standard error of prediction reported was 0.2 g/L as epicatechin equivalents, with a residual predictive deviation (RPD) of 2.97. Total anthocyanin content (ACY) and total phenolic index (TPI) were also estimated, using UV-Vis prediction models with standard errors of prediction of 24.69 mg/L malvidine-3-glucoside equivalent and 1.2 for the anthocyanins and TPI. RPD values for the prediction models were 7.12 (anthocyanins) and 13.15 (TPI).

Oenological parameters such as pH, titratable acidity, volatile acidity, alcohol content, glycerol, glucose, and fructose were measured with a WineScanTM FT120 instrument (Foss Electric, Hillerød, Denmark) [39]. 

The HPLC analysis for individual phenolic compounds was performed with a diode array HPLC detector, adapted from a method developed by Peng et al. (2002) [40]. An Agilent Technologies 1260 Infinity series (Agilent, Waldbronn, Germany) HPLC system was used for the determination of the individual wines’ phenolic composition. Orthophosphoric acid in deionized water at 1.5% and acetonitrile (100%) were used as solvent A and B, respectively. The gradient elution flow rate was established as follows: 0 min (5% solvent B), 73 min (25% solvent B), 78 min (50% solvent B), 86 min (50% solvent B), 90 min (5% solvent B), with a flow rate of 1 mL/min and 15 min of re-equilibration time between injections. A reversed-phased column (PLRP-S polymeric) with 3 µm particle size, 100 Å pore size, and dimensions of 150 mm × 4.6 mm was used at 35 °C. Compounds were detected using a photodiode array detector after the injection of 20 µL of sample. Gallic acid and monomeric and polymeric flavan-3-ols were quantified at 280 nm. The 320 nm, 360 nm, and 520 nm wavelengths were used to quantify the individual phenolic acids, flavonols, and anthocyanins and polymeric pigments, respectively. Some of the individual phenolic compounds (gallic acid, catechin, caffeic acid, p-coumaric acid, quercetin-3-glucoside, quercetin, and malvidin-3-glucoside) were quantified using external calibration, while the remaining compounds with no available external calibration were quantified using the calibration of compounds that corresponded to the same phenolic family (e.g., malvidine-3-glucoside for the anthocyanins).

### 2.4. Sensory Evaluation

#### 2.4.1. Trained Panel (T)

The trained panel consisted of 13 panelists with little experience in wine evaluation (10 females and 3 males, ranging in ages from 29 to 68 years old, with an average of 45.16), who were recruited based on their availability and interest. They participated in weekly one-hour sessions over a period of 17 weeks. The training consisted of 15 sessions, and the formal tastings were performed during the last two weeks. Panelists were paid for their participation.

The panelists read and signed written consent forms in which they were informed about the samples that they would taste during the sessions. They were told that they would participate in a project about red wine mouthfeel perception, but they were not informed about the scientific hypothesis of the study.

During the training, the panelists learned about basic in-mouth attributes (sweet, sour, bitter, astringent, and burning sensation). The training process began with a theoretical description of human taste and a figure that illustrated the in-mouth perception of the mentioned attributes. The next five sessions consisted of the recognition and ranking of the different attributes. For this, panelists were presented with reference standards for each attribute. For sweetness, standard solutions of fructose at 1, 4, 6, and 8 g/L were used. Tartaric acid solutions at 0.5, 1.5, 3, and 6 g/L were used for sourness training. Quinine sulphate solutions were presented to explain bitterness at different levels: 0.0075, 0.015, and 0.035 g/L. Solutions with aluminum sulphate (0.6, 1.5, 2.5, and 5 g/L) and commercial tannins (0.2, 0.6, 1.0, and 2.0 g/L) represented different intensities of astringency. Finally, ethanol was used to prepare solutions that illustrated different intensities of burning sensation (7%, 12%, and 15% *v*/*v*). All standard solutions were prepared in both water and model wine solution.

The following three sessions consisted of familiarization with the scale and discussions to reach consensus on perceived intensities. For this, a commercial red wine was used as a wine base and spiked with the standards at the different levels. Panelists rated the perceived intensity of each attribute using a 10 cm unstructured line scale ranging from 0 (absence of the descriptor) to 100 (extremely high).

As the panelists struggled to reach consensus on the intensity of the perceived astringency, the next four sessions were focused especially on this specific attribute. During these sessions, different commercial wines elaborated with the use of various winemaking techniques and of different aging periods were used to represent a wide range of astringency, from very low to very high. These wines were subsequently used for rating and discussion by the panelists. At the same time, panelists also evaluated the base wines, spiked with commercial tannins at different levels (0.2 and 0.6 g/L), to further understand the ranges of astringency perceived in wine. 

During the last training session, the panelists got familiarized with the sensory software, participating in a tasting simulation. This session was considered an important step to prevent the panelists from being biased during the formal tasting by any doubt or technical issue. Panelists were asked to taste different commercial wines, and they were shown how to use Compusense *five* software (Compusense *five* version 5.5, Guelph, Ontario, Canada). 

In each training session, one sample was presented in duplicate to monitor panelist reproducibility. In addition, consensus was achieved during the training sessions through discussions in order to minimize the carry-over effects between samples. The panelists agreed that the use of a cleanser (pectin solution at 1 g/L) and a three-minute break helped them to better discriminate between the samples, especially for the evaluation of astringency. 

#### 2.4.2. Expert Panel (E)

Ten panelists (three females and seven males from 24 to 66 years old, with an average age of 37.5 years) were recruited based on their interest and availability. Considering the criteria defined by Parr et al. (2004) [36], the recruited panelists were considered experts as they all were established winemakers. The experts also signed a consent form and were informed of the fact that they would taste red wine for mouthfeel attributes, without being informed about the real aim of the project. They were not paid for their participation.

Before the formal tasting sessions, the experts participated in a one-hour discussion session. The experts were presented with wines containing different levels of standards for sweet (2 and 6 g/L fructose), sour (0.5 and 3 g/L tartaric acid), bitter (0.0075 and 0.03 g/L quinine sulphate), astringent (0.2 and 0.6 g/L commercial tannins), and burning sensation (12% and 15% *v*/*v* ethanol). During the session, the experts familiarized themselves with the scale (the same as was used for the trained panel) and the use of the pectin solution as a mouth cleaner. In addition, a discussion was held, followed by achieving consensus on the importance of the break between samples.

#### 2.4.3. Formal Tastings

The study was performed on two consecutive days, and both panels followed the same tasting protocol. Each day, panelists participated in two tastings, separated by a 13-minute break. During these sessions, the wines were presented in black ISO (International Organization for Standardization) glasses [41] labeled with a three-digit code and covered with petri dishes. Using a William Latin square design generated by Compusense *five* software, each panelist received the samples in a different order. 

The panelists received 30 mL of each sample at room temperature (20 ± 2 °C). They were asked to rate the sweetness, sourness, bitterness, astringency, and burning sensation using the unstructured line scale, as described above. They were instructed to use the sip and spit protocol between samples by using water and pectin solution (1 g/L) [42]. A forced three-minute break was also taken between samples.

The tastings were organized in such a way as to not have more than 12 wines tasted in one session. This was because panelists felt that the wines were very difficult to taste and evaluate. The in-mouth basic perception of the wines was in a very similar sensory space, and they felt that this was the maximum number that they could deal with. Therefore, two tasting sessions were organized during two consecutive days. Each session consisted of two tastings separated by a 13-minute break, and the total of four sessions were organized as follows.

Three of the four tastings were termed “varietal” (V) and consisted of presenting wine samples from the same variety (Pinotage, Shiraz, or Cabernet Sauvignon) at a time. During these “varietal” tastings, the panelists received five samples consisting of the four wines of each specific variety, plus a blind replicate to evaluate the panelist’s consistency and sample replication. 

The fourth tasting was termed “mixed” (M) and consisted of presenting the judges with a set of six wines. For this tasting, the 12 wines were divided into two sets of six wines each (two wines from each grape variety, representing low and high MCP tannin levels, respectively). Set 1 consisted of the NEK***P***, SB***P***, WER***S***, SOO***S***, NEJ***CS***, and SL***CS*** wines, and Set 2 consisted of the PV***CS***, WP***CS***, NEJ***P***, PV***P***, PV***S***, and WEG***S*** wines. These codes used during the study were generated from the abbreviations of the winery´s name and italic bolded letters stands for the grape cultivar (***CS*** for Cabernet Sauvignon, ***P*** for Pinotage and ***S*** for Shiraz). Half of the panelists thus evaluated one of the sets and the other half evaluated the other set of six wines.

In addition, the set of wines was randomized to ensure the minimization of variety or tasting modality bias. 

Compusense *five* Sensory Software was used for the instructions, scales, and data collection.

### 2.5. Statistical Procedures

The panel performances were compared considering different criteria: Panelist consistency (repeatability) and consensus (agreement with other judges), as well as the ability to discriminate between samples and attributes. To check panelist repeatability and consensus, Spearman tests were used, and the correspondent intraclass correlation coefficients (ICCs) were calculated (ICC agreement and ICC consistency). The intraclass correlation coefficient is the ratio of the variance of the real rating scores over the sum of the variance of the true ratings and the variance of error.

Analysis of variance (ANOVA), principal component analysis, and the RV coefficient were used to identify similarities between the two panels regarding product discrimination. The significance of the RV coefficients was tested using a permutation test. In order to check sample or attribute discrimination, paired comparisons of the means were carried out using the least significant difference test (*a* = 5%). Correlations between attributes, wine, and chemical composition were calculated by means of multifactorial analysis (MFA). A combination of Statistica V13 (StatSoft, Inc.; Tulsa, OK; USA) and R (https://www.r-project.org/) was used to perform the statistical analyses.

## 3. Results and Discussion

As observed in Table 1, the tannin levels of the wines ranged from 285 to 2568 mg/L, with variations in their anthocyanin levels and total phenol index, within the range of concentrations reported in the literature [23]. All the wines were dry red wines with very low levels of sugars, an alcohol content higher than 13%, and pH, titratable acidity, and volatile acidity levels within the range normally described in these types of wines.

Performance parameter values for both panels are provided in Table 2. Intrapanel agreement and consistency were highly relative to other attributes for astringency in the case of both panels (ICC (agreement) T = 0.733, ICC (consistency) T = 0.718, and ICC (agreement) E = 0.799, ICC (consistency) E = 0.833 for the trained (T) and expert panels (E), respectively), indicating the ability of the judges to accurately discriminate this attribute. Generally, the expert panel seemed to have a better agreement and repeatability for all the described attributes. This may be due to the experts’ better analytical skills in discriminating between attributes, especially in the case of such a complex sensation as astringency, usually confused with sourness or bitterness by judges [26]. However, even if the intraclass coefficients were higher for the expert panel, ANOVA analysis indicated no statistically significant differences between the two panels, showing a similar performance between both panels.

Interpanel agreement and consistency were highest for astringency (ICC (agreement) T_E = 0.608 and ICC (consistency) T_E = 0.601), with all the other attributes having low (close to zero) reliability values. These results showed that both panels might have adopted similar analytical strategies when evaluating this sensation. Both panels perceived the intensity of astringency in a similar way, but not the other parameters, which may have been due to the sample itself. From the scores obtained for sweet, sour, bitter, and burning sensation (Table 3 and Table 4), it can be observed that the difference between the samples with the lowest intensity and the samples with the highest intensity very rarely exceeded two points on the 10-point scale. The differences between wines when evaluating these attributes were not very clear, varying with the panel and the tasting modality. Furthermore, in order to reach a better understanding of our results, the effect of the panel, as well as the possible interactions panel*modality and panel*grape variety, were statistically investigated.

Generally, both panels scored the wines similarly, and the only significant panel effect was found for sweetness, where the expert panel scored significantly higher than the trained panel (*F* = 4.66, *p* = 0.05).

When investigating whether the ratings were significantly affected by the tasting modality or not, the attributes scores were analyzed with a three-way ANOVA, including as factors or variables the panelist group (trained panel, expert panel), tasting modality (varietal, mixed), and wine samples. With a 95% level of confidence, the results showed that independently of the tasting modality, the two panels evaluated similarly the wines (*F_sweet_* = 0.75, *p* = 0.39; *F_sour_* = 2.64, *p* = 0.12; *F_bitter_* = 0.20, *p* = 0.67; *F_astringent_* = 0.63, *p* = 0.44; and *F_burning_sensation_* = 0.83, *p* = 0.37). Moreover, the possible interaction wine cultivar and the two panels were also checked by performing a three-way ANOVA with panelist group (trained panel, expert panel), wine variety (Pinotage, Shiraz, and Cabernet Sauvignon), and wine as factors. The interactions indicated that the previous results where no differences were found between the two panels were not influenced by wine cultivar. Burning sensation did show a trend for an interaction, but post hoc tests still indicated no difference between E and T (*F_burning_sensation_* = 2.88, *p* = 0.06). 

In most cases, both panels rated the sweetness as low (Table 3 and Table 4), and this was expected due to all the samples being dry red wines with low sugar levels (Table 1). This might also have led to the difficulty in discriminating among the samples, therefore resulting in low intrapanel consistency and consensus. Considering the number of statistically significant differences found between the samples, we could evaluate the discrimination abilities that the panel had for this specific attribute. For the trained panel, only two significant differences were found between samples: The Pinotage PV***P*** wines were less sweet than the Pinotage NEK***P*** wines in the “varietal” tasting modality and the Shiraz WEG***S*** wines in the “mixed” tasting modality. The experts found that the Shiraz PV***S*** wines were less sweet than the Shiraz SOO***S*** wines for the “varietal” modality, and for the “mixed” modality the Cabernet Sauvignon SL***CS*** wines were perceived as sweeter than the Shiraz SOO***S*** wines for Set 1. Meanwhile, the Pinotage NEJ***P*** wines were sweeter than the Cabernet Sauvignons PV***CS***, the Shiraz PV***S***, and the Shiraz SOO***S*** wines for Set 2. Even though the agreement and consistency were not very good for sweetness, the expert panel seemed to have better discrimination skills for this attribute (more significant differences between samples compared to the trained panel), especially when they received the wines in the “mixed” modality.

Sourness was also rated very similarly among all the wines, except for the SOO***S*** wines, which obtained the lowest ratings from the trained panel when presented in the “varietal” modality (Table 3). The similarity of the wines for sourness was not surprising, as the differences of the wines in titratable acidity was also small, with a maximum variation of 0.81 g/L tartaric acid equivalents between the lowest (SOO***S*** wines) and the highest (PV***P*** wines) levels of the wines (Table 1). These small differences in sourness intensity could be one of the reasons for the lower performance of the trained panel when rating this attribute (Table 2). However, the experts tended to have better performance for this attribute, given that, even if the differences were small, the experts had the ability to separate this attribute from the rest and give a more accurate evaluation.

In the case of bitterness, whereas the trained panelists were not able to significantly discriminate between the samples in either of the two tasting modalities, the experts showed better discrimination in both tasting modalities (Table 3 and Table 4), especially for Cabernet Sauvignon and Shiraz wines. Independently of the tasting modality, the experts generally perceived the SOO***S*** wine as the sample with the highest bitterness. 

When it came to astringency, both panels used the same part of the scale to evaluate the wines, scoring for the less astringent with means around 3.5 and for the most astringent wines with means higher than 7.5. However, when looking at the sample discrimination abilities for this attribute (number of significant differences between samples by ANOVA, *p* < 0.05), few significant differences were found between samples, and this was only found during the “varietal” tasting modality. The trained panelists perceived the PV***S***, WEG***S***, and WER***S*** wines as less astringent than the SOO***S*** wines when presented in the varietal modality, whereas the expert panel only perceived the NEJ***P*** wines as less astringent than the SB***P*** wines. 

Concerning the burning sensation, the experts performed better than the trained panelists (number of significant differences between samples by ANOVA, *p* < 0.05). When tasting in the “varietal” modality for Pinotage, the SB***P*** and NEK***P*** wines were perceived to have a higher burning sensation than the NEJ***P*** and PV***P*** wines. For Shiraz, the WEG***S*** wines were perceived to have a lower burning sensation than the SOO***S*** and WER***S*** wines. For the “mixed” modality tasting, the experts perceived NEJ***CS*** as being less intense than NEK***P*** and SOO***S***. 

These results, and the good interpanel agreement for astringency, create the impression that both panels used similar analytical strategies to evaluate astringency. However, the trained panelists discriminated better for this sensation. Due to the training sessions, their analytical skills might have improved, and they might have shared a better consensus on the concept of astringency. Generally, it is difficult to evaluate this sensation separately, because sourness and bitterness are often confused with astringency [26]. Therefore, it is important to also have a good ability to discriminate the rest of the attributes when differences are quite small among samples, as was the case in our study. The fact that the experts had a better discrimination criterion when it came to separate and evaluate individually these wine attributes seemed to make them a more suitable choice in this type of study. 

However, in order to compare the two panels, it was also essential to assess performance for the overall wine evaluation. A good way to evaluate the global perception of the same samples by two panels is to compare the similarity of the configuration of sample distribution in the sensory space. This can be quantified by the RV coefficient [43], and it has previously been shown to be useful when the intention is to derive configurations comparing sensory profiling and free sorting [35,44,45]. These coefficients can be computed when PCA is realized and can provide information on panelist agreement on sample differentiation. RV coefficients range between 0 and 1, with values closer to 1 indicating greater similarity.

Figure 1A–D represent the sample configurations given by the first two components of the PCA, which were obtained from the sensory scores of two panels for the two different tasting modalities. The explained variances (more than 70%) were very similar among the four PCA maps, but the sample distribution varied according to the panel and the tasting modality. From the different sample distributions, it can be observed that, globally, experts discriminated better among samples, especially for the “varietal” modality (Figure 1B). These results confirmed the expectations from the ANOVA and agree with other authors. On the one hand, when comparing a trained panel to experts, Zamora and Guirao (2004) [46] found that experts had better discriminatory abilities when profiling Chardonnay wines, and Arvisenet et al. (2016) [30] showed that experts and trained panelists had similar behaviors when studying taste–aroma interactions in model wine. On the other hand, when comparing two trained panels, Canul et al. (2011) [45] showed that the panel that was more familiar with the product due to exposure was more discriminative and repetitive when evaluating fresh “Cuajada” cheese. 

When evaluating the intrapanel similarity between the configurations of the samples obtained by the two panels for the two tasting modalities, the RV coefficient computed for the trained panel was 0.18 (comparing Figure 1A to Figure 1C), lower than the 0.40 computed for the expert panel (comparing Figure 1B to Figure 1D). Moreover, this result shows that the agreement between trained panelists decreased when changing the modality from "varietal" to "mixed", which means that they were not able to discriminate the samples similarly in both situations. This result indicates that the tasting modality may have influenced the evaluation of the samples, but, as mentioned previously, no significant effect was found for any of the attributes. During the "mixed" modality, not all the wines were presented to all the panelists. It may be that the combination of the wines itself in each set might have introduced bias by not being different enough. As was shown previously in the discussion (Table 4), few significant differences between samples were found for Set 2. 

With the aim of checking interpanel similarity, the configurations given by each panel within each tasting modality (Figure 1A,B for “varietal” and Figure 1C,D for “mixed”) were compared. Similarities between panels varied with the tasting modality, with RV coefficients of 0.52 for the "varietal" and 0.39 for the "mixed" modality. However, the "varietal" modality seemed more adequate to use for sample presentation in order to obtain better consensus between the two panels when discriminating between wines.

Up until now, univariate analysis (ANOVA), as well as multivariate analysis (PCA), have indicated that the sample discrimination could vary with the panel and with the tasting modality. This may be an issue when one wants to explain sensory perceptions through the chemical composition of the product, as it gives rise to a question: Does the variability in the ratings, due to panel or tasting modality, affect the sensory–chemical correlations?

To answer this question, a multifactor analysis (MFA) was performed, including the sensory evaluations from the two panels and both tasting situations (Table 3 and Table 4), the oenological parameters (Table 1), and the phenolic composition of the 12 wines (Appendix A) used for the study. Figure 2 displays the map of computed Pearson’s correlations. To investigate the possible existence of any risk of misinterpreting the sensory–chemistry relationship, correlations between the sensory attributes and the chemical parameters were studied. As can be observed in Figure 2, the scores for sensory attributes, except for bitterness and astringency, were not always correlated in the same way with the chemistry data. These correlations showed a variation according to panel and tasting modality. In general, sourness and bitterness were negatively correlated (*r* = −0.61, *p* = 0.03), whereas factors such as bitterness and astringency were positively correlated (*r* = 0.79, *p* < 0.01). 

First, the given scores for the wine sweetness were positively correlated between panels when tasting in the "varietal" modality (*r* = 0.59, *p* = 0.04), and significant negative correlations were shown for cyanidine-3-glucoside (*r* (E) = −0.66 and *r* (T) = −0.58, *p* < 0.05). However, the panel seemed to have an impact on the correlations between this attribute and the different chemical parameters, whereas the sweetness perceived by the experts was positively correlated with the presence of grape reaction product (GRP) (*r* = 0.81) and fructose (*r* = 0.59), and negatively with kaempferol (*r* = −0.78). It showed a positive correlation with caffeic acid (*r* = 0.60) and a negative correlation with peonidin-3-glucoside (*r* = −0.59), quercetin (*r* = −0.64), and total anthocyanin (*r* = −0.70) when described by the trained panel.

Secondly, the sourness seemed to be highly influenced by both the tasting modality and the panel. Positive correlations were found between the ratings given by the two panels in the "varietal" modality (*r* = 0.73, *p* < 0.05), but not in the "mixed" modality. When analyzing the "varietal” modality, sourness showed a positive correlation with the total acidity for both panels (*r* (E) = 0.72 and *r* (T) = 0.69, *p* < 0.05). However, when evaluating the "mixed" modality, only the scores given by the experts showed this positive correlation between sourness and the total acidity (*r* (E) = 0.68, *p* < 0.05). These results showed a better ability to discriminate sourness from other attributes by the experts compared to the trained panel. Nevertheless, and as expected, sourness was negatively correlated with fructose or glucose. It has been previously shown that sugar levels can diminish the perception of the acidity of wines [44]. However, in this case also, other correlations became significant, except for the trained panel in the “mixed” modality. 

In the case of bitterness, the sensory scores given by the two panels were positively correlated with the astringency scores. The correlations between these attributes and the chemical composition varied with the tasting modality and the panel. Some common correlations between the panels were shown for the tannin levels (*r* (E) = 0.67 and *r* (T) = 0.63, *p* < 0.05) and total phenolic content (*r* (E) = *r* (T) = 0.60, *p* < 0.05) in the “varietal” modality, as well as for peonidin-3-coumarylglucoside (*r* (E) = 0.66 and *r* (T) = 0.60, *p* < 0.05) in the “mixed” modality. However, the correlations between the “varietal” tasting scores and the chemical composition were closer to the scientific expectations, as it has been shown previously that tannin and phenolic compounds play an important role in the bitter sensation of wines [45].

Although the level of astringency was positively correlated between the panels, the *r*^2^ values varied between the tasting modality, showing a 0.86 for “varietal” and a 0.78 for “mixed”. These correlations confirmed the good interpanel agreement shown in Table 1. In terms of sensory and chemical correlation, a positive correlation was generally found between the perceived astringency, MCP tannins, polymeric phenols, and total phenolics for both panels. These results agreed with what has been described in the literature [47,48]. Looking into the tasting modality, we found a greater correlation from both panels between astringency and these chemistry parameters for the “varietal” modality. Therefore, the sensory scores given by the experts and the trained panelists in the “varietal” modality were highly correlated with the levels of tannins and total phenols, with *r*^2^ values of 0.93 and 0.89 and 0.86 and 0.87, respectively. However, when correlating with the results from the “mixed” modality, the *r*^2^ values were lower: 0.65 and 0.71 for the experts, and 0.42 and 0.37 for the trained panel, with the latter correlations not being statistically significant. These results indicate that the tasting modality might have possibly biased the astringency rating, especially for the trained panelists, who already had shown less discriminatory ability when changing modality. However, as previously mentioned, these effects were not found to be statistically significant in the present study.

Depending on the tasting modality, different correlations were found between astringency and the rest of the chemical parameters. On the one hand, when the judges evaluated the wine in the “varietal” modality, astringency also showed a positive correlation with gallic acid (*r* (E) = 0.69, *r* (T) = 0.61), coutaric acid (*r* (E) = 0.73, *r* (T) = 0.71) and the ethanol of the wines (*r* (E) = 0.61, *r* (T) = 0.61). On the other hand, in the “mixed” modality, astringency was also positively correlated with the polymeric pigments (*r* (E) = 0.60, *r* (T) = 0.66). Apart from the differences found between the two tasting modalities, the different panels also influenced these correlations. For example, for the “varietal” modality, the astringency as measured by the experts was correlated with catechin (*r* (E) = 0.66) and quercetin-3-glucoside (*r* (E) = 0.62), whereas the astringency as measured by the trained panelists was positively correlated with polymeric pigments (*r* (T) = 0.67, *p* < 0.05), pH (*r* (T) = 0.64, *p* < 0.05), and glycerol (*r* (T) = 0.58, *p* < 0.05). In the case of the “mixed” modality, astringency as measured by the trained panelists was also correlated with glycerol (*r* (T) = 0.60), and for the experts’ correlations were found with polymeric phenols (*r* (E) = 0.69) and quercetin-3-glucoside (*r* (E) = 0.62). However, independently of the panel and the modality, the general expectation that higher levels of tannins or phenolic compounds would be translated into higher astringency was confirmed. 

Finally, when considering the burning sensation, one would expect a positive correlation with the ethanol content [44], but only the scores given by the experts in the “varietal” modality confirmed this expectation (*r* = 0.62). In the case of the experts, this sensation was also positively correlated with volatile acidity (*r_varietal_* = 0.71 and *r_mixed_* = 0.77) for both modalities and negatively correlated with cyanidin-3-acetylglucoside (*r* = −0.59) for the “varietal” modality. For the answers given by the trained panel, no correlations were shared between the two tasting modalities. However, for the “varietal” modality, some significant correlations were found with gallic acid (*r* = 0.67), coutaric acid (*r* = 0.75), and total acidity (*r* = −0.69). Within the “mixed” modality, a positive correlation was found with pH (*r* = 0.66) and GRP by both panels (*r* (E) = 0.88 and *r* (T) = 0.62).

Finally, the answer to the question, “Does the variability in the ratings, due to panel or tasting modality, affect the sensory-chemical correlations?” seems to be affirmative in the case of our study. Only the correlation of the astringency with the tannins, polymeric phenols, and total phenol index seemed to not be affected by the panel and tasting modality, probably because both panels, trained panelists and experts, had a shared common idea for astringency and this was reflected into good intra- and interpanel performance. The tasting modality seemed to have the greatest impact. “Varietal” modality seemed to show more stable correlations between panels. The fact that the relationships between sensory perception and chemistry were weaker when tasting in the “mixed” modality indicates that presenting wines from different varieties at the same time might bias the sensory perception. This may be due to the differences between the sensory spaces other than mouthfeel provided by the grape variety itself. Even though caution was taken during this study by using a cleanser and a forced break between samples, it seems that the simultaneous presentation of the wines coming from different grape varieties might have caused lower consistency among the judges. This could have been due to the fact that the two combinations of samples for this modality were chosen randomly, resulting in samples with very different sensory profiles that could induce fatigue due to carry-over effects. Another cause could have been the fact that the samples used for the study were chosen based only on tannin levels, and no screening of sensory space discrimination was performed, with the risk that differences between the wines might be too difficult to be perceived and accurately evaluated.

## 4. Conclusions

This study showed that both a trained panel and experts were able to accurately evaluate the same red wines and to reach consensus. Among the attributes, astringency played the most discriminative role, whereas small differences were detected between the samples for the rest of the attributes. The trained panel demonstrated better discriminative abilities for the astringency but struggled to accurately discriminate the samples for the rest of the attributes. The experts showed good discriminative abilities for all the attributes, even when the differences between samples were small. Additionally, they showed very similar responses for the wine astringency when compared to the results obtained from the trained panel.

The selection of the wines may have had an impact on the results of the study. The wines were chosen to cover a wide range of astringency looking at their tannin levels, but less attention was paid to their differences in sweetness, sourness, bitterness, and burning sensation. However, in our case, the samples were chosen with the aim of covering different astringency levels and, even without tasting them during the screening process, this aim was fulfilled. 

It is risky to draw a conclusion regarding the absence of a tasting modality effect based on this study, as the number of wines and tasters for each combination was low. The tasting modality seemed to affect the sensory responses and, even if it was not shown to be statistically significant, caution should be taken when deciding sample presentation. Further research to study the effect of the tasting modality should be done, including a more complete experimental design with more wine combinations and tasters.

Good agreement was observed between the panels when tasting wines in the varietal modality, with more stable correlations between chemistry data and sensory scores. The results from this study show that the use of experts instead of trained panelists would be a time- and cost-effective option when these types of studies are planned. The experienced winemakers showed similar performance to the trained panel after only a one-hour discussion session before the tasting and were able to better discriminate samples with small differences.

## Figures and Tables

**Figure 1 foods-08-00003-f001:**
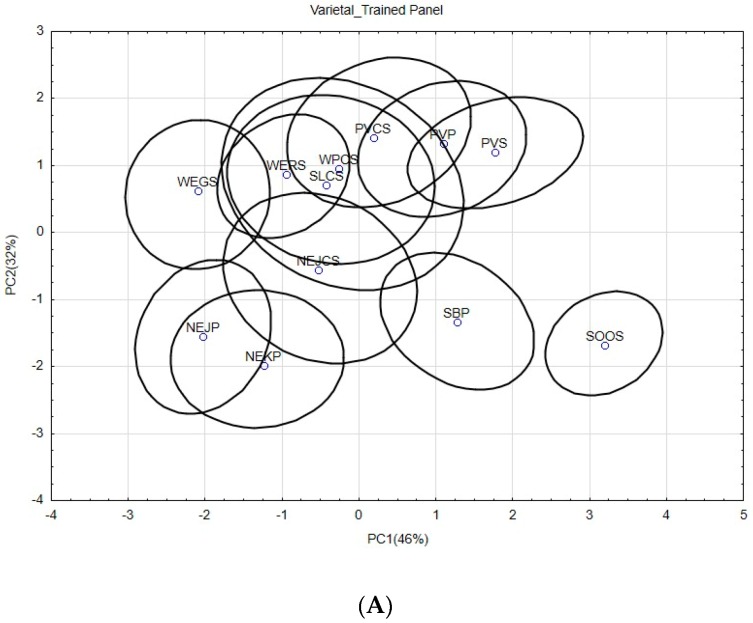
Principal component analysis performed on the sensory variables given by the two panels in the two tasting modalities. (**A**) Configuration of the sample distribution as obtained from the scores given by the trained panel within the “varietal” tasting modality; (**B**) configuration of the sample distribution as obtained from the scores given by the expert panel within the “varietal” tasting modality; (**C**) configuration of the sample distribution as obtained from the scores given by the trained panel within the “mixed” tasting modality; (**D**) configuration of the sample distribution as obtained from the scores given by the expert panel within the “mixed” tasting modality.

**Figure 2 foods-08-00003-f002:**
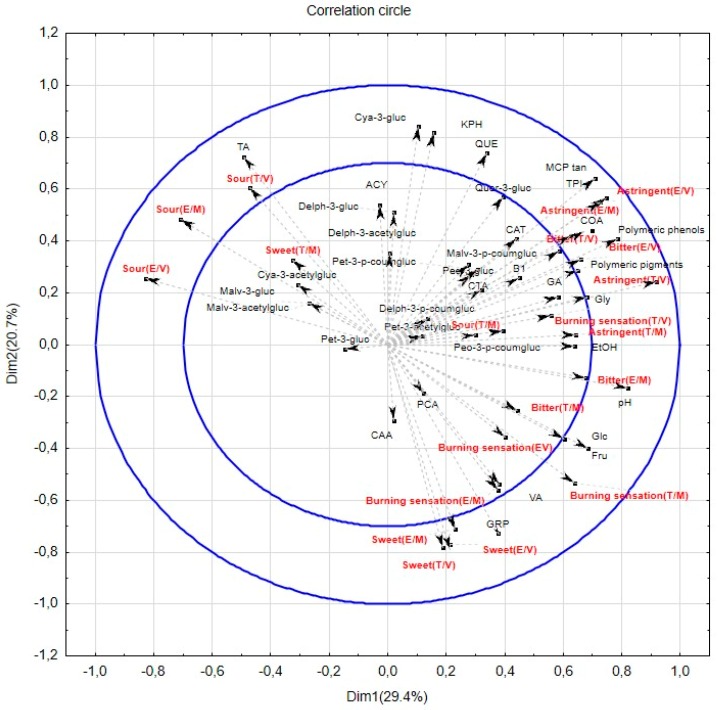
Correlation circle of multifactor analysis performed on the sensory variables given by the two panels in the two tasting modalities and the physicochemical composition of the wine samples. The sensory variables were sweet, sour, bitter, astringent, and burning sensation. The physicochemical variables were volatile acidity (VA), total acid (TA), pH, ethanol concentration (EtOH), MCP tannin levels (MCP tan), anthocyanins (ANT), total polyphenol index (TPI), glucose (Glc), fructose (Fru), glycerol (GLY), gallic acid (GA), grape reaction product (GRP), delphinidin-3-glucoside, caftaric acid (CTA), cyanidin-3-glucoside, (+)-catechin (CAT), procyanidin B1 (B1), petunidin-3-gluc, caffeic acid (CAA), coutaric acid (COA), peonidin-3-glucoside, malvidin-3-glucoside, delphinidin-3-acetylglucoside, cyanidin-3-acetylglucoside, p-coumaric acid (pCA), petunidin-3-acetylglucoside, quercetin-3-glucoside, malvidin-3-acetylglucoside, delphinidin-3-coumarylglucoside, malvidin-3-coumarylglucoside, polymeric pigments, quercetin (QUE), and kaempferol (KPH).

**Table 1 foods-08-00003-t001:** Oenological parameters of the wines.

Wine	Variety	pH	VA ^a^	TA ^b^	Ethanol (*v*/*v* %)	MCP tan ^c^	ACY ^d^	TPI ^e^	Glucose (g/L)	Fructose (g/L)	Glycerol (g/L)
NEJ***CS*** ^f^	Cabernet Sauvignon	3.6	0.50	6.2	14.86	2118	487.8	73.59	0.39	1.22	12.41
PV***CS***	Cabernet Sauvignon	3.6	0.43	6.2	14.73	1389	447.0	54.67	n.d.	1.15	10.81
SL***CS***	Cabernet Sauvignon	3.8	0.47	5.8	14.95	1541	425.2	58.35	0.03	2.58	11.85
WP***CS***	Cabernet Sauvignon	3.7	0.57	6.0	14.79	2297	523.7	72.36	n.d. ^g^	1.51	11.96
NEJ***P***	Pinotage	3.5	0.64	5.8	14.33	896.9	462.5	47.66	0.78	1.33	11.25
NEK***P***	Pinotage	3.6	0.87	5.9	14.95	967.3	481.8	49.06	0.81	1.81	11.29
PV***P***	Pinotage	3.5	0.56	6.2	14.36	2045	776.6	78.86	0.54	1.28	10.93
SB***P***	Pinotage	3.7	0.73	5.6	14.77	1716	569.8	63.99	0.86	1.48	11.07
PV***S***	Shiraz	3.6	0.73	6.1	15.08	2567	640.5	84.64	0.87	1.25	11.26
SOO***S***	Shiraz	4.0	0.72	5.4	15.04	2485	631.0	83.73	1.08	3.30	11.89
WEG***S***	Shiraz	3.5	0.58	6.0	13.17	285.1	514.3	38.68	0.03	0.97	10.64
WER***S***	Shiraz	3.6	0.64	5.8	14.76	622.8	700.6	51.04	0.27	1.24	10.81

^a^ Volatile acidity expressed as g/L acetic acid. ^b^ Titratable acidity expressed as g/L tartaric acid. ^c^ Tannin levels expressed as mg/L epicatechin equivalents. ^d^ Total anthocyanin levels expressed as mg/L malvidine-3-glucoside equivalents. ^e^ Total polyphenol index expressed as absorbance units (AU). ^f^ Codes used during the study were generated from the abbreviations of the winery´s name and italic bolded letters stands for the grape cultivar (***CS*** for Cabernet Sauvignon, ***P*** for Pinotage and ***S*** for Shiraz). ^g^ n.d.: not detected. VA: volatile acidity; TA: total acid; MCP: methyl cellulose precipitable; ACY: total anthocyanin content; TPI: total polyphenol index.

**Table 2 foods-08-00003-t002:** Intrapanel reliability results.

Panel	Attribute	ICC (Agreement)	ICC (Consistency)	Correlation	*p*-Value	SEM
T	Sweet	0.105	0.097	−0.120	0.700	0.430
	Sour	−0.055	−0.051	0.020	0.950	0.788
	Bitter	0.116	0.109	0.400	0.200	0.844
	Astringent	0.733	0.718	0.720	0.010	0.636
	Burning sensation	0.127	0.118	0.120	0.720	0.631
E	Sweet	0.201	0.188	0.200	0.540	0.662
	Sour	0.279	0.262	0.720	0.010	0.763
	Bitter	0.302	0.287	0.540	0.070	0.600
	Astringent	0.799	0.833	0.920	0.000	0.401
	Burning sensation	0.558	0.556	0.550	0.060	0.433
T vs. E	Sweet	−0.078	−0.146	−0.260	0.130	0.859
	Sour	−0.148	−0.147	−0.180	0.300	0.867
	Bitter	−0.138	−0.150	−0.020	0.910	0.954
	Astringent	0.608	0.601	0.670	0.000	0.710
	Burning sensation	0.228	0.326	0.180	0.300	0.580

ICC (consistency) and ICC (agreement): intra-class correlation coefficients; correlation: Spearman correlation; *p*-value: test for significant correlation; SEM: standard error of measurement; T: trained panel; E: expert panel.

**Table 3 foods-08-00003-t003:** Means, standard deviation, and results of analysis of variance (ANOVA) for the two panels in the “varietal” tasting modality.

Grape Variety	Wine Name	Sweet		Sour		Bitter		Astringent		Burning Sensation	
		T ^a^	E ^b^	T	E	T	E	T	E	T	E
**Cabernet Sauvignon**	NEJ***CS*** ^e^	1.21 ^c^ (1.11 ^d^)	1.41 (1.20)	3.64 (1.71)	3.97 (2.33)	2.98 (2.01)	4.69 (1.68)	6.11 (1.96)	5.82 (2.16)	3.26 (1.60)	3.92 (2.30)
	PV***CS***	1.28 (1.08)	2.03 (1.70)	4.34 (1.48)	4.35 (1.64)	3.81 (1.98)	4.20 (1.68)	4.43 (2.11)	5.52 (1.73)	3.71 (1.97)	3.96 (2.20)
	SL***CS***	1.53 (1.58)	2.67 (1.95)	4.32 (1.97)	3.29 (1.88)	3.83 (2.34)	3.52 (1.61)	4.81 (2.47)	5.45 (1.61)	3.22 (2.50)	3.83 (1.79)
	WP***CS***	0.90 (0.68)	2.07 (1.73)	3.94 (1.75)	3.68 (1.63)	3.44 (2.09)	4.78 (1.90)	6.00 (2.02)	6.20 (1.81)	3.15 (2.11)	4.02 (2.39)
**Pinotage**	NEJ***P***	1.58 (1.57)	2.94 (2.18) **	3.43 (1.71)	4.52 (1.85)	2.96 (2.25)	3.47 (1.83)	4.53 (1.93)	4.45 (1.95)	2.83 (1.35)	3.95 (2.23)
	NEK***P***	1.84 (1.87)	3.02 (1.99)	3.48 (1.73)	3.82 (1.90)	3.09 (1.75)	3.96 (1.76)	4.95 (2.09)	5.11 (1.77)	3.09 (1.40)	5.28 (1.85) **
	PV***P***	0.98 (1.07)	2.49 (1.72) *	4.00 (1.65)	4.28 (2.18)	3.90 (2.17)	4.15 (1.90)	5.76 (2.16)	5.93 (1.82)	3.69 (1.84)	3.93 (2.00)
	SB***P***	1.54 (1.90)	2.83 (1.93)	3.37 (2.06)	3.20 (2.11)	3.52 (2.49)	4.28 (2.60)	6.31 (1.89)	5.71 (1.72)	3.82 (1.81)	5.13 (2.24)
**Shiraz**	PV***S***	1.18 (1.14)	2.08 (1.30)	4.16 (2.26)	3.51 (1.99)	4.17 (2.31)	3.88 (2.81)	6.52 (2.02)	6.76 (2.03)	3.70 (1.90)	4.58 (2.06)
	SOO***S***	1.46 (1.61)	3.31 (2.62) *	2.90 (1.52)	2.86 (1.94)	4.12 (1.93)	4.65 (2.00)	7.18 (1.52)	5.98 (1.56)	4.14 (1.39)	4.01 (2.16)
	WEG***S***	1.06 (0.90)	2.29 (1.70) **	3.86 (2.11)	4.09 (1.70)	2.96 (1.98)	3.15 (1.79)	3.47 (2.01)	4.00 (1.94)	3.16 (1.81)	3.08 (1.69)
	WER***S***	1.08 (1.11)	2.72 (1.53) *	3.88 (1.30)	3.81 (1.45)	3.41 (1.86)	3.25 (1.48)	3.90 (1.50)	3.98 (2.00)	3.38 (1.68)	4.11 (1.89)

^a^ T: Trained panel; ^b^ E: Expert panel; ^c^ mean value; ^d^ standard deviation; ^e^ Codes used during the study were generated from the abbreviations of the winery´s name and italic bolded letters stands for the grape cultivar (***CS*** for Cabernet Sauvignon, ***P*** for Pinotage and ***S*** for Shiraz); * statistical significance between T and E for *p* < 0.05; and ** statistical significance between T and E for *p* < 0.01.

**Table 4 foods-08-00003-t004:** Means, standard deviation, and results of ANOVA for the two panels in the “mixed” tasting modality.

Wine Set	Wine Name	Sweet		Sour		Bitter		Astringent		Burning Sensation	
		T ^a^	E ^b^	T	E	T	E	T	E	T	E
**Set 1**	NEJ***CS*** ^e^	0.97 ^c^ (1.28) ^d^	2.70 (2.04)	4.65 (1.33)	3.12 (1.68)	3.48 (1.67)	3.86 (1.60)	7.33 (1.83)	6.60 (1.70)	3.68 (1.42)	3.10 (2.21)
	NEK***P***	1.17 (1.87)	3.24 (2.54) *	4.25 (1.35)	3.26 (1.78)	3.30 (2.35)	3.66 (2.02)	4.70 (1.91)	5.04 (1.64)	4.00 (1.52)	5.06 (2.78)
	SB***P***	0.65 (0.63)	3.24 (1.42) *	3.92 (1.32)	2.62 (1.78)	3.60 (2.09)	3.70 (2.17)	6.40 (1.60)	5.54 (2.02)	3.82 (1.49)	4.06 (1.59)
	SL***CS***	1.35 (0.90)	3.38 (1.790) *	4.53 (1.27)	3.46 (1.46)	3.73 (2.24)	3.54 (1.81)	6.45 (1.35)	5.30 (2.00)	3.62 (1.32)	3.82 (1.99)
	SOO***S***	1.18 (2.04)	2.36 (2.07)	4.32 (2.20)	2.90 (1.20)	4.45 (2.54)	5.52 (2.22)	7.53 (1.67)	6.26 (1.56)	4.32 (2.45)	5.00 (0.46)
	WER***S***	0.92 (1.63)	2.44 (1.85)	4.10 (1.21)	3.68 (0.97)	3.97 (2.14)	3.22 (1.99)	4.53 (1.77)	3.68 (1.87)	3.92 (1.88)	4.12 (2.49)
**Set 2**	NEJ***P***	2.36 (1.16)	2.58 (1.97)	3.50 (2.14)	3.54 (2.16)	2.74 (1.93)	3.64 (1.86)	3.69 (1.98)	3.52 (2.01)	3.49 (1.85)	3.74 (2.79)
	PV***CS***	1.54 (0.72)	1.02 (0.41)	3.70 (1.29)	4.22 (2.11)	3.24 (1.88)	3.30 (2.21)	4.30 (2.14)	4.12 (1.71)	3.26 (2.48)	3.80 (2.02)
	PV***P***	1.99 (0.85)	1.32 (0.73)	4.86 (1.61)	3.86 (1.91)	3.34 (2.55)	3.26 (2.05)	4.60 (2.30)	5.68 (2.55)	3.66 (1.49)	3.98 (3.09)
	PV***S***	1.54 (0.81)	1.42 (0.73)	3.59 (2.00)	3.56 (2.17)	3.23 (2.32)	4.50 (2.68)	5.39 (1.93)	6.50 (2.60)	3.37 (1.72)	4.38 (3.49)
	WEG***S***	1.19 (0.84)	1.62 (1.04)	3.79 (1.94)	3.88 (2.30)	3.89 (3.16)	4.20 (2.68)	6.31 (2.19)	6.00 (2.83)	3.31 (2.12)	4.16 (2.98)
	WP***CS***	1.67 (1.00)	2.02 (2.63)	3.66 (1.76)	3.66 (1.83)	3.30 (2.91)	3.58 (1.81)	6.13 (1.55)	6.18 (2.56)	3.66 (1.87)	3.58 (2.45)

^a^ T: Trained panel; ^b^ E: Expert panel; ^c^ mean value; ^d^ standard deviation; ^e^ Codes used during the study were generated from the abbreviations of the winery´s name and italic bolded letters stands for the grape cultivar (***CS*** for Cabernet Sauvignon, ***P*** for Pinotage and ***S*** for Shiraz); * statistical significance between T and E for *p* < 0.05.

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
