# Peer review of "Basic In-Mouth Attribute Evaluation: A Comparison of Two Panels"

_foods, 2018, doi:10.3390/foods8010003_

Round 1
Reviewer 1 Report
From my point of view the work merits to be published in Foods. The paper is well written and the topic is of interest.
I have a minor suggestion concerning presentation of results. From my point of view the data related to the correlation of sensory and chemical data, that are included in discussion, should remain in results as they are mainly descriptive. Thus I suggest to make a joint "results and discussion" section including the text that now is in sections "results" as well as "discussion".
Reviewer 2 Report
this work relies on panel performance of trained panel versus experts. in the title,I believe that "basic in-mouth attributes as predicted by the phenolic composition of red wines..." can be properly changed in " basic in-mouth attribute evaluation: a comparison of two panels" , it is more appropriate since "the prediction" is more based on maths regression.
L40: references refer to the influence of matrix in model solution and not in wine, so is better to specify or to add works that evaluated the influence of ethanol....on astringency in real conditions
L45-48: several methods involving precipitation (with ovalbumin, saliva..) and well correlated to astringency have been developed..it is better to illustrate all of them and then the method used in the work and why.
L90 is not clear how many wines?please explain
L184: wines were tasted in duplicate?or not?. in this case the fewer assessors explain the variability in data
L273-286. please better explain the difference of perception between the two modalities of tasting
in fig 1-2 I think is better to not use ellipse, but points to better visualize samples
L364:if RV value is between 0 and 1, the 0.18 seem to refers to dissimilarity
L371 and L453: the sentence refers to discussion, but this is the result section..
this part is very confusing L371-377 seems to be part of conclusions, while
in discussion there are only results without real discussion, in fact, MFA is a result. in addiiton the explained variance of mFA is very low! can author better explain this point?
and as stated previuosly sweetness was the only attribute not in accordance intra-panel, please explain why now in L413-420 is correlated with these attrbute.
this part about correlation is confusing and there are no discussion neither references on correlation between polyphenols and sensory attributes. please better explain and present data and add relative discussion, trying to be more concise with less hypotesis.
Reviewer 3 Report
Results and Discussion sections can be combined and somewhat truncated. I also don't think that having anything in bold in the abstract is a good idea.
Round 2
Reviewer 2 Report
thanks for answers, now I feel that the paper is improved and more clear in result and discussion section. for clarity I suggest to add some references to sentence L42-43. for the rest, I'm ok